# Human Sperm Chromosomes: To Form Hairpin-Loops, Or Not to Form Hairpin-Loops, That Is the Question

**DOI:** 10.3390/genes10070504

**Published:** 2019-07-03

**Authors:** Dimitrios Ioannou, Helen G. Tempest

**Affiliations:** 1Department of Human and Molecular Genetics, Herbert Wertheim College of Medicine, Florida International University, Miami, FL 33199, USA; 2Biomolecular Sciences Institute, Florida International University, Miami, FL 33199, USA

**Keywords:** nuclear organization, chromatin, spermatozoa, chromosomes, chromosome territories, centromeres

## Abstract

Background: Genomes are non-randomly organized within the interphase nucleus; and spermatozoa are proposed to have a unique hairpin-loop configuration, which has been hypothesized to be critical for the ordered exodus of the paternal genome following fertilization. Recent studies suggest that the hairpin-loop model of sperm chromatin organization is more segmentally organized. The purpose of this study is to examine the 3D organization and hairpin-loop configurations of chromosomes in human spermatozoa. Methods: Three-color sperm-fluorescence in-situ hybridization was utilized against the centromeres, and chromosome p- and q-arms of eight chromosomes from five normozoospermic donors. Wide-field fluorescence microscopy and 3D modelling established the radial organization and hairpin-loop chromosome configurations in spermatozoa. Results: All chromosomes possessed reproducible non-random radial organization (*p* < 0.05) and formed discrete hairpin-loop configurations. However, chromosomes preferentially formed narrow or wide hairpin-loops. We did not find evidence to support the existence of a centralized chromocenter(s) with centromeres being more peripherally localized than one or both of their respective chromosome arms. Conclusion: This provides further evidence to support a more segmental organization of chromatin in the human sperm nucleus. This may be of significance for fertilization and early embryogenesis as specific genomic regions are likely to be exposed, remodeled, and activated first, following fertilization.

## 1. Introduction

### 1.1. Organization of the Human Genome

The human genome consists of 23 pairs of chromosomes, each chromosome pair is in essence a unique linear sequence of DNA nucleotides that differs in length, with chromosome 1 being the longest (~250 Mbp), and chromosome 21 being the shortest (~48 Mbp). The linear sequence of our genome is housed in the cell nucleus (~10 micrometers). If DNA were to be unraveled, and disassociated from various proteins it would reach almost 2 meters in length. Somehow, the considerable amount of information contained in DNA must be efficiently packaged so that the genome can not only fit within the nucleus, but also in a way that facilitates a myriad of required normal cellular functions (e.g., DNA transcription, DNA replication, DNA damage recognition and repair etc.) [1]. So, how is our genome packaged in a functional manner? If asked to describe the packaging of DNA many would describe what is most often portrayed in textbooks. This classical view often depicts each linear sequence of DNA being efficiently packaged by wrapping itself around histones to form a nucleosome, which is then packaged to form a solenoid, which is further packaged into various loops and domains. During the metaphase stage of cell division, DNA is further packaged and maximally condensed into highly organized structures, known as chromosomes. Frequently, (particularly for cytogeneticists), when we think about these linear strands of DNA the mental picture conjured up is that of a chromosome with its classical features including the telomeres, centromere, and the short (p) and long (q) arms. However, it is important to recognize that the metaphase chromosome only exists for a very short period of the cell cycle. So what happens to the packaging and organization of these linear pieces of DNA during the remainder of the cell cycle when it is more decondensed? 

The genome even during interphase, when the DNA is at its most decondensed, remains highly organized with chromosomes forming distinct chromosome territories (CTs). The existence of CTs is not a new concept and was in fact, first proposed over 110 years ago by Theodore Boveri following his studies of blastomeres in horse roundworms [2]. However, the concept of CTs was largely forgotten until the work of Thomas Cremer and others in the 1970s and 1980s (reviewed in [3,4,5]). The existence and organization of CTs in the interphase nucleus has been experimentally established for many decades and widely accepted in scientific literature [6]. Nevertheless, the majority of texts still erroneously omit the existence of CTs when depicting the structure and packaging of DNA. Many studies have utilized fluorescence in-situ hybridization (FISH) to study the spatial organization of CTs. The majority of these studies have demonstrated that CTs exhibit distinct radial patterns of nuclear organization in somatic cells, which has been observed in two- and three-dimensions (2D and 3D). Of note, somatic chromatin organization has been shown to be cell-type specific [5], evolutionarily conserved [7], and reproducible between individuals [8,9,10,11]. Distinct radial patterns of CT organization have been reported for the majority of CTs in diverse cell-types and species. Radial CT organization typically follows the gene density or chromosome size model, in which CTs are preferentially radially organized based on either of these chromosome characteristics [12]. The unique organization of CTs has been postulated to serve as an additional layer of epigenetic regulation of the genome that likely plays an important role in normal cellular functions [13]. Largely, the radial organization of the majority of CTs in diverse cell-types and species typically follows one of the two models previously described. However, there is one notable exception, spermatozoa, which seemingly displays a unique model of organization, compared to other cell types. Given that spermatozoa are ellipsoid in shape and contain a flagellum, some organization studies have assessed radial and/or longitudinal organization of chromosomes or targeted loci [10,11,14,15,16,17,18,19,20,21,22,23,24,25,26,27,28]. The vast majority of these studies have reported non-random chromatin organization in sperm from various species, with the exception of chicken [26,27]. The majority of studies that have assessed the radial organization of chromosomes in sperm, suggest that the organization tends to follow the gene density model [11,16,17,22]. However, a distinct pattern of organization in sperm was proposed by the Zalensky group in the 1990s—the hairpin-loop model. This model described chromosome centromeres clustering in the center of the nucleus to form a single or multiple chromocenters, and the p- and q-arms of each chromosome lying in parallel with each other, or intermingled as they stretched from the center of the nucleus to the nuclear periphery where the telomeres formed dimers and tetramers (Figure 1A) [25,29,30,31,32]. This unique CT configuration was originally described by the Zalensky group as the hairpin-loop model, due to its resemblance of a hairpin-loop structure. The description of the hairpin-loop model led to this term being adopted and routinely utilized in the field to describe sperm chromatin organization. If the analogy of a bicycle wheel is used the centromeres would form the hub, the chromosome arms the spokes, and the telomeres the wheel rim. However, few studies have revisited this model of sperm chromatin organization since it was proposed over 20 years ago. Recently, we published a study that reexamined this model of organization utilizing both 2D and 3D approaches to assess the organization of centromeres and telomeres in normozoospermic males. The study findings led us to conclude that the original hairpin-loop model needed to be further refined. We provided interindividual reproducible evidence of multiple chromocenters that were not localized centrally and telomeres that were not restricted to the nuclear periphery as previously suggested, but rather centromeres and telomeres were localized throughout the nucleus. Thus, we have suggested that sperm chromatin organization is perhaps less like the bicycle wheel (Figure 1A) but rather more segmentally organized in the nucleus, with hairpin-loop structures forming in different directions and orientations in the nucleus (Figure 1B) [6,10]. 

### 1.2. Why Do Sperm Exhibit A Different Organization to Somatic Cells?

The sperm cell is a highly specialized cell, whose main function is to transport safely the paternal genome during its long, arduous journey to the oocyte. The process of spermatogenesis is a unique process whereby spermatogonial stem cells produce diploid spermatogonia which ultimately undergo meiosis and differentiate to form four haploid spermatozoa. During spermiogenesis, some of the most dramatic transformations in cell biology take place. Haploid round spermatids undergo substantial remodeling and repackaging of the genome into one of the smallest cells; a streamlined spermatozoon, which also contains an acrosome cap, a mid-piece enriched with mitochondria and a flagellum [33]. Despite the efficient packaging of the genome in somatic cells, spermatozoa have evolved an even more effective mechanism through which to condense and package its genome. In humans, the majority (85–95%) of DNA associated histones are initially replaced with transition proteins and finally with protamines [34], with the remaining 5–15% of the genome retaining histone packaging [35]. The largely protamine packaged sperm genome has been proposed to serve several critical functions. These may include: (i) enabling a smaller cell volume and facilitating the hydrodynamic shape required to fulfill its function; (ii) protecting the paternal genome from DNA damage, as sperm lack DNA repair mechanisms; and (iii) deprogramming and inactivating the genome resulting in a relatively inert “silent” vessel [34,36]. Given the unique features of the sperm cell, it is perhaps not surprising that CTs display a unique spatial organization when compared to somatic cells. 

### 1.3. What Could Be the Functional Consequence of Chromatin Organization in Spermatozoa?

Increasingly, evidence suggests that the sperm cell may not in fact be a “silent vessel”; but rather the sperm cell delivers an epigenetically primed genome to the maternal oocyte. Elegantly designed studies have demonstrated that histone bound regions of the paternal genome are more decondensed and are preferentially enriched in specific genes and gene promoters (including imprinted genes, developmentally important signaling proteins and transcription factors etc.) [20,37,38,39,40]. Furthermore, DNA hypomethylation of developmentally important gene families in the sperm cell may permit early transcriptional activation during early embryogenesis [34,38]. Recent high resolution molecular studies in mice have demonstrated that the transcriptional repressor CTCF binding motif is preferentially associated with the more condensed protamine packaged regions of the mouse genome [41]. Hi-C studies evaluating 3D interactions in murine gametes and early embryogenesis have identified distinct patterns between male and female gametes and embryos, with sperm containing more associations and long-range interactions than oocytes [42,43]. The unique, hierarchical CT organization throughout spermatogenesis is likely to be important for the creation of, and normal function of spermatozoa. However, many have proposed that CT organization in sperm is an integral aspect of epigenetic mechanisms required for early embryogenesis [6,13,44,45]. During fertilization, the epigenetically primed paternal genome is hypothesized to be gradually exposed and remodeled by the oocyte [23,31]. Thus, many have also proposed that sperm CT organization is an additional level of epigenetic programming that may play a crucial role in the formation of the male pronucleus and early embryonic development [6,18,29,31]. These findings suggest that we need to revisit the concept of spermatozoa being “silent vessels”, rather spermatozoa carry an epigenetically primed paternal genome, which not only delivers critical information to the oocyte, but may be poised to activate critical genes for early embryogenesis following fertilization [41]. Furthermore, the CT organization most likely serves as another critical aspect of the epigenetically poised sperm nucleus. CT organization likely functions as a unique mechanism to deliver, unpackage, remodel, and transfer the genetic and epigenetic information from the paternal genome to the oocyte, perhaps in a highly ordered fashion [46]. Perhaps specific chromosomes or chromosomal regions need to be delivered sequentially to the maternal ooplasm to respond to oocyte signals to begin the formation of the male pronucleus [23]. Therefore, CT organization in the sperm nucleus may play a more important role in early embryonic divisions than currently perceived. Given the potential functional significance and unique CT organization in spermatozoa, we wanted to extend our initial 3D study of the organization of centromeres and telomeres to evaluate centromeres and the p- and q-arms of multiple chromosomes to establish whether we could confirm the existence of chromosome hairpin-loop formations. This was of particular interest given our newly proposed more segmentally organized sperm nucleus, which was based on centromere and telomere organization, and did not assess chromosome arm configuration and organization [6,10]. 

## 2. Materials and Methods

### 2.1. Patient Cohort and Semen Analysis

This study was conducted in accordance with the Declaration of Helsinki, and the protocol was approved by the Ethics Committee of Florida International University (FIU-IRB-13-0044). All methods were carried out in accordance with the approved guidelines. Cryopreserved semen samples were purchased from the Xytex Cryo International sperm bank. Five sperm donors of proven fertility were included in this study. Semen samples were collected via masturbation; and were classified as normozoospermic based on the World Health Organization criteria [47] semen parameter guidelines. 

### 2.2. Semen Sample Preparation

Semen samples were prepared for FISH by removing the seminal fluid as described in detail previously [11]. In brief, cryopreserved semen samples were thawed at room temperature and washed with sperm wash buffer (10 mM NaCl, 10 mM Tris, pH 7.0) followed by centrifugation for 7 min at 504 g. Subsequently, the supernatant was removed and the pellet was resuspended with fresh sperm wash buffer and centrifuged as noted previously. Semen samples underwent a further 3–5 cycles of removal of the supernatant, addition of fresh wash buffer and centrifugation depending on the pellet size. Following the last sperm wash the supernatant was removed and the sample was then fixed drop-wise using 3:1 methanol:acetic acid to a final volume of 5 mL, the sample was then centrifuged as previously described and the fixation steps were repeated depending on the pellet size 3-5 times. Following fixation washes the pellet was resuspended in 1–1.5 mL of fixative depending on the pellet size and 1–3 µl of the sample was spread onto a glass slide. Slides were evaluated under a differential interference contrast microscope (Olympus BX53) for optimal cell density prior to initiation of FISH.

### 2.3. Sperm-FISH

The same FISH approach as described previously was applied in this study [10]. In brief, spermatozoa were spread onto glass microscope slides at the optimum density to minimize overlapping nuclei, cells then underwent a mild formaldehyde fixation to maintain the 3D structure of the nuclei as much as possible. This fixation step was followed by mild decondensation of sperm nuclei to facilitate FISH probe access in the densely protamine packaged sperm nuclei. This was achieved through a 20 min incubation in decondensation buffer (10 mM DTT (Sigma Aldrich, St Louis, MO, USA), 10 mM Tris solution, pH 8.0) in the dark. Following decondensation slides were rinsed in 2 × saline sodium citrate (SSC; Fisher Scientific, Pittsburgh, PA, USA), before dehydrating through an ethanol series (70–80–100%). Three-color FISH was performed for eight chromosomes, chromosomes: 2, 3, 6, 8, 10, 12, 16, and 18. FISH probe targets for each chromosome included the chromosome short (p) and long (q) arms, and satellite enumeration probes against the centromeres (Figure 2A). These specific chromosomes were chosen to evaluate a range of chromosome sizes and gene density; and to include both metacentric and submetacentric chromosomes, whilst excluding acrocentric chromosomes, which lack p-arms. Critically these specific chromosomes were also selected as centromere probes for these chromosomes do not cross hybridize with other centromeres. For each of the investigated chromosomes, arm paints were labelled in green (p-arms) and orange (q-arms) fluorochromes (MetaSystems, Boston, MA, USA) and centromeres were labelled with aqua fluorochromes (Kreatech, distributed by Leica Biosystems Buffalo Grove, IL, USA). FISH probes were utilized as per the manufacturer’s guidelines. A 1:1:1 ratio mix of all 3 probes was denatured at 73 °C for 10 min and subsequently co-denatured with sperm cells for 90 s in a Thermobrite^®^ Statspin (Abbott Molecular, Illinois, IL, USA) followed by hybridization for a minimum of 16 h at 37 °C. Post hybridization washes were carried out as per the manufacturers guidelines with the addition of a ddH_2_O rinse for 1 min and an ethanol series (70–80–100%). Following these washes, slides were air dried, and subsequently mounted with 4’,6-diamidino-2-phenylindole (DAPI) antifade mounting medium (Vector Labs, Burlingame, CA, USA) under a 24 × 55 mm coverslip. 

### 2.4. 3D FISH Image Acquisition, Image Rendering and Analysis

The same methodology as previously reported was utilized to capture and render images in 3D [10]. Cells were imaged utilizing the DeltaVision high-resolution widefield fluorescence microscope (GE Healthcare Life Sciences, Pittsburgh, PA, USA) system; consisting of an Olympus IX71 inverted microscope with 60X, 1.4 NA oil-immersion lens and a photometric CCD. All images were taken with a Z step size of 0.2 μm (92 optical sections), saved as 3D stacks and subjected to constrained iterative deconvolution using the same standard settings (DeltaVision–SoftWoRx -V 5.5; GE Healthcare Life Sciences, Pittsburgh, PA, USA) (Figure 2A). A minimum of 30 images per subject, per probe set were acquired using TRITC (594 nm), FITC (523 nm), CFP (480 nm) filters. 3D stacks from SoftWoRx were reconstructed, rendered in 3D and analyzed utilizing Imaris software (V.7.6.3 Bitplane–Zurich, Switzerland) by converting images to 32-bit float images. The nuclear periphery was established using the DAPI counterstain and was rendered by creating a surface in Imaris. Similarly, the targeted loci (centromeres, p- and q-arms) were established based on fluorescence intensity of the FISH probes and translation of the pixel intensity resulted in a rendered surface (Figure 2B). To establish the nuclear periphery and model the FISH probes, an iso-surface was created to visualize the object in 3D space, whilst allowing verification of the accuracy directly against the original raw image. The creation of the nuclear surface was defined by setting an intensity threshold to select the voxels that were considered part of the reconstructed iso-surface. Minimal manual user intervention was required to establish the threshold, to best reflect the raw data set, and remove any background fluorescence. Voxel selection was further enhanced by applying a Gaussian filter prior to selection, to remove the noise not attributed to the labelled cell. This smoothing step adjusts for the limits of resolution of the acquisition system, and quality of the tissue labelling. This step ensures that the voxels from "out of focus light", which appear as a background blur in the Z-plane, were not included as a part of the surface structure ensuring a more accurate representation of the sperm nucleus. The reconstructed DAPI surface was overlaid on the raw image to ensure the created surfaces fitted the raw data in all three axes (X, Y, and Z). To measure the radial nuclear localization of target loci within the nucleus, the DAPI surface object was utilized to denote the nuclear periphery, this was used as a region of interest to isolate the other fluorescent channels (FISH signals) within the nucleus. The masking process generated a new channel based on the voxels that were located inside of the 3D volume of nuclear periphery for each individual fluorochrome. This new channel was rendered without interference from other voxels in the dataset, creating a new surface segmentation of structures within the nucleus. The Imaris Distance Transform (DT) tool utilizes a 3D quasi-Euclidean distance transformation from the geometric center (Figure 2C) of each rendered FISH signal in the data set to the binary mask of the DAPI border of the surface (nuclear periphery). The Imaris DT tool calculates the shortest distance in 3D space between each data point (FISH signals) and the DAPI nuclear periphery (surface border) with minimal user intervention, following the contours of the nucleus [10,48,49,50]. The Imaris DT tool is shape invariant and does not impose any assumptions on the cell shape or size [51]. To facilitate comparisons across experiments, it is desirable to have a measure that is both scale, and shape invariant; thus Imaris DT measurements were normalized against the widest “radial” diameter of individual sperm nuclei to account for differences between individual nuclei. The radius of each nuclei was divided by three to create three distinct nuclear regions (interior, intermediate, or peripheral). The radial organization of each FISH probe was assessed by measuring the distance of the geometric center of each loci to the nearest nuclear periphery as measured by the Imaris DT tool. The position of the geometric center of the FISH probe was then assigned to one of the three radial regions (interior, intermediate, or periphery). Hairpin-loop chromosome configurations were established by identifying the geometric center (Figure 2C) of each targeted loci (p-arm, q-arm, and the centromere) and determining the angle created between the p- and q-arms through the centromere (Figure 2D). Based on the angle created by the p- and q-arms, hairpin-loop configurations were arbitrarily stratified into two categories, those forming angles less than or equal to 40°, or greater than 40° to evaluate whether chromosomes had a tendency to form narrower or wider hairpin-loop configurations. The unique morphology of the sperm cell often facilitates assessment of longitudinal organization. Unfortunately, in this study the levels of background fluorescence were low, thus, rarely was the sperm tail visible (< 10% of analyzed cells) (Figure 2A), which precluded robust assessment of the longitudinal organization of the targeted loci. Thus, longitudinal organization or assessment of hairpin-loop configurations in relation to the sperm tail was not possible in the current study.

### 2.5. Statistical Analysis

The Chi-squared goodness of fit (χ2) was utilized to evaluate if the radial organization of each target of interest differed from random, a *p*-value of < 0.05 suggested a non-random distribution. In essence, if the target loci were equally distributed across the three radial segments of the nucleus it suggested a random distribution, whereas preferential distribution in one or more segments as determined by the Chi-squared goodness of fit test indicated non-random organization of the target loci.

## 3. Results

In this study, we examined the 3D radial organization and chromosome configurations of eight different chromosomes in five normozoospermic subjects. A total of 1240 sperm were analyzed in this study, with a minimum of 30 nuclei evaluate per subject, per target loci (centromeres, p- and q-arms). Decondensation of nuclei should be avoided if at all possible when assessing the nuclear organization of chromatin. However, due to the unique protamine packaging in sperm, it is unfortunately a prerequisite for sperm-FISH. As published previously [10], we optimized the decondensation conditions to ensure efficient FISH hybridization (>95%) and minimal reproducible swelling between experiments and samples. A formaldehyde fixation step was performed prior to decondensation to maintain the 3D nuclear structure as much as possible. The decondensation parameters utilized in this study, resulted in a mild reproducible decondensation resulting in an average 2 fold increase in DAPI volume in decondensed sperm nuclei versus native sperm nuclei. Additionally, the increased nuclear volume in this study is comparable with, or lower than those previously reported in the literature [14,29]. 

### 3.1. Radial Organization of Chromosome Centromeres, p- and q-Arms in Sperm Nuclei

The radial organization of the geometric center of the three target loci (centromeres, p- and q-arms) for each of the eight investigated chromosomes (2, 3, 6, 8, 10, 12, 16, and 18) was evaluated by measuring the micrometer distance from the geometric center of each target to the nearest nuclear edge (Figure 3). The radius of the nucleus was utilized to normalize the distance measured for the geometric center of each probe target. The radius of each individual nucleus was divided into three regions (peripheral, intermediate, and interior) and the geometric center for each target loci was assigned to one of these three regions based on the distance from the nuclear periphery. All investigated chromosome targets were found to be non-randomly organized (*p* < 0.05) in the sperm nuclei from all five subjects with the exception of chromosome 12 centromere, which was found to be randomly organized (*p* > 0.05) (Figure 3C). Assessing the radial distribution of the three target loci for each chromosome in the three nuclear domains (interior, intermediate and periphery) several patterns emerge. The localization of the q-arms in each of the three nuclear regions is similar for all investigated chromosomes, ranging from ~17–24% found in the interior region, ~51–61% in the intermediate region, and ~20–25% in the nuclear periphery (Figure 3A). For the p-arms, a similar pattern of radial organization is observed for the majority of investigated chromosomes, ranging from ~16-31% found in the interior region, ~47–59% in the intermediate region, and ~19–29% in the nuclear periphery. However, the p-arms of chromosome 10 and 16 were localized more (~31%, chromosome 10) or less frequently (~16%, chromosome 16) in the nuclear interior in comparison to the other p-arms (Figure 3B). The radial positioning of the centromeres of the tested chromosomes appears much more variable when compared to the p- and q-arms, ranging from ~15–42% in the interior, ~34–61% in the intermediate region, and 18–29% in the nuclear periphery (Figure 3C). The radial position of centromeres 2, 3, 6, 8, 12, and 16 account for most of the variation observed particularly in the interior and intermediate regions of the nucleus. 

To establish whether we could provide evidence of centralized chromocenters in human sperm nuclei, we utilized the micrometer measurements from the geometric center of each target loci to the nearest nuclear edge (Table 1). None of the investigated chromosomes strongly supported the concept of a centralized chromocenter, whereby one would expect centromeres to be more centrally localized than their respective chromosome arms (Table 1). When analyzing the raw data from the 155 cells studied in the five subjects, reproducible radial patterns of organization emerged for each the investigated loci. The data from the 1240 individual sperm cells analyzed revealed that centromeres were preferentially more distally localized in the sperm nucleus when compared to their respective chromosome arms, with 30.5% of centromeres being more distally located than either the p- or q-arm, and 37.8% being more distally located than both p- and q-arms (data not shown). Thus, the centromeres were found to be more peripherally localized in the sperm nucleus than at least one chromosome arm more than two-thirds of the time. Looking at the mean data presented in Table 1 from the five subjects, five out of the eight investigated centromeres (chromosomes 3, 6, 12, 16, and 18) were more peripherally localized than both the p- and q-arms of the respective chromosomes. Additionally, the centromeres for chromosomes 2, 8, and 10 were more distally localized than the q-arm but not the p-arm of the respective chromosomes; albeit the localization of the centromere, p- and q-arm is very similar for chromosome 2. 

### 3.2. Chromosome p- and q-Arm Configurations in Sperm Nuclei; Is A Hairpin-Loop Formed?

The configurations of each chromosome were assessed by measuring the angle formed by the p- and q-arms through the centromere for each chromosome in the five subjects enrolled in this study. Visualization of the 3D organization of the p- and q-arms of each individual chromosome in sperm nuclei clearly revealed that the p- and q-arms formed distinct territories with little, to no, evidence of intermingling between the two chromosome arms in the vast majority of nuclei (Figure 1 and Figure 2). For the investigated chromosomes we provide evidence to partially support the hairpin-loop model of chromosome organization in human sperm. Our findings clearly demonstrate that the chromosome arms (p and q) had a tendency to lie in parallel to one another forming what can be described as a hairpin-loop configuration (Figure 1 and Figure 2). Analysis of the angles formed by the p- and q-arms through the centromere revealed that when looking at the average of the 155 cells analyzed in the five subjects for each chromosome, the majority of spermatozoa had chromosome hairpin-loop configurations that were less than or equal to 80°. We stratified chromosomes based on the percentage of cells that contained hairpin-loop configurations that were greater than 80°. Between 7–16% of the configurations of chromosomes 2, 8, 10, 12, and 18 were greater than 80°, whereas between 25–38% of the configurations for chromosomes 3, 6, and 16 were greater than 80°. Given that for a significant proportion of chromosomes, the hairpin-loop configurations observed were less than 80°, we further stratified the chromosome configurations to less than or equal to 40° and greater than 40° (Figure 4). The data obtained in this study demonstrates that individual chromosomes preferentially form narrower or wider hairpin-loop configurations that were similar between the five subjects enrolled in this study. The data in Figure 4 displays the chromosomes in order of narrowest (chromosome 10) to widest (chromosome 16) hairpin-loop configuration based on the average data from the five subjects enrolled in this study. 

Additionally, correlations between chromosome size, gene density and centromere localization and type of hairpin-loop configurations (narrow or wide) were examined. Analysis of the data presented in Figure 4 shows no obvious correlation with between narrow or wide hairpin-loop configuration and chromosome size. For example, chromosome 16 is the second smallest investigated chromosome and preferentially exhibits a wide hairpin-loop configuration, whereas chromosome 18 is the smallest chromosome, and preferentially exhibits a narrow hairpin-loop configuration. Using the Ensembl genome browser correlations between chromosome gene density and hairpin-loop configurations could also be assessed. Chromosome length and number of coding genes for each of the investigated chromosomes was obtained from Ensembl (genome assembly GRCh38.p12). No clear correlation between gene density and tendency to form narrow or wide hairpin loop configurations was observed. For example taking into account the chromosome length and number of protein coding genes, chromosomes 16, 12, 8, 6, 10, 3, 2, and 18 can be organized from highest to lowest gene density respectively. Similar to chromosome size no discernable correlation was observed for gene density and hairpin-loop configuration. For example, looking at the three chromosomes with the highest gene density, they were ranked as third, fifth, and eighth widest hairpin-loop for chromosomes 12, 8, and 16 respectively. Similarly, no correlation was observed between preference to form narrow or wide hairpin-loop configuration with distance of the centromere to the nuclear edge.

## 4. Discussion

In this study, we investigated the 3D nuclear organization of three major chromosome components (centromeres, p- and q-arms), and how these were configured and formed CTs in sperm nuclei. We targeted eight different metacentric or sub-metacentric chromosomes of varying size and gene density in five normozoospermic males. The results of this study provides further evidence of a non-random radial organization of chromosomes in sperm nuclei, which was reproducible between the enrolled subjects. Of the 24 loci evaluated, all but one (chromosome 12 centromere) were found to be non-randomly radially organized in human sperm. This is not the first study to report the non-random organization of CTs or other target loci; with multiple studies reporting similar observations in humans and evolutionarily divergent species [10,11,14,15,16,17,18,19,20,21,22,23,24,25,26,28,32]. Thus, the finding of non-random radial organization for the centromeres, p- and q-arms for the eight investigated chromosomes, is perhaps not surprising. Nevertheless, albeit in a small sample size, one of the major strengths of the current study is that we have examined the radial organization of the various loci between subjects and report a largely reproducible pattern of organization. Interindividual differences are rarely examined in nuclear organization studies, however, a handful of published also report similar findings [10,11,15,19].

CT organization in spermatozoa is hypothesized to possess a unique hairpin-loop structure, which differs from other cell types. The hairpin-loop model proposes that the centromeres form a single or multiple chromocenter in the nuclear interior, with the p- and q-arms of the chromosomes stretching out toward the nuclear periphery where the telomeres form dimers and tetramers [29,31,52]. It is important to note that one of the original experimental papers examining the hairpin-loop conformation in sperm utilized FISH probes against the telomeres, p- and q-arms solely and inferred the position or localization of the centromere based on overlapping signals between the p- and q-arms [29]. The inference of centromere localization is challenging given that this study reported intertwisted spirally-coiled structures and overlapping regions which could lead to erroneous assignment of centromere localization. In the current study, we utilized FISH probes against not only the p- and q-arms but also the centromeres of each investigated chromosome to ensure the CT conformation and localization could be accurately determined for the chromosome arms and centromeres. In support of the previous study, we observed that the p- and q-arms did in fact form a configuration that reflected a hairpin-loop structure, providing further evidence to support the sperm hairpin-loop model of CT organization [25,29,30,31,32]. In the current study, the conformation of the p- and q-arms in sperm nuclei were observed to form discrete separate territories with virtually no overlapping or intermingling of the territories for all of the investigated chromosomes, except in rare instances (Figure 1 and Figure 2). Similar findings have also been reported in earlier stages of spermatogenesis [53]; in human lymphocytes and amniotic cells [54] with observed intermingling being limited to a narrow boundary zone. These findings contradict an earlier study that reported the p- and q-arms of three investigated chromosomes (1, 2, and 5) to be tightly bound resulting in intertwisted spirally-coiled structures or closely aligned parallel to each other [29]. Additionally, we observed that certain chromosomes preferentially formed narrow or wide hairpin-loop configurations as determined by the angle created by the arms of each chromosome through the centromere. However, there was no obvious correlation between the angle formed and chromosome size, gene density, or centromere position. The preference for certain chromosomes to form hairpin-loops that had a tendency to be narrower (e.g., chromosomes 10 and 18) or wider (e.g., chromosome 6 and 16) was largely reproducible between the enrolled subjects, suggesting that this hairpin-loop configuration may be functionally significant.

Despite the aforementioned data, results from this study only partially supports the hairpin-loop model of sperm chromosome organization; given that we were not able to find evidence to support the chromocenter aspect of the model. Rather data from the current study further supports our hypothesis to suggest that the sperm nucleus is reproducibly more segmentally organized than initially hypothesized [6,10]. In line with our previous findings, we observed centromeres to be localized throughout the nucleus (interior, intermediate, and periphery), not just restricted to the nuclear interior. The majority of centromeres (> 65%) were found to be more peripherally localized than one or both of the p- and/or q-arms. These findings are supported by an earlier study that also noted centromeres to be more peripherally organized during earlier stages of spermatogenesis [53]. Of note, the eight chromosome centromeres investigated exhibited the most variation in terms of preferential radial distribution in the three nuclear regions when compared to the chromosome p- and q-arms. In the current study, we established the radial organization based on the localization of the geometric center of each target loci as established by the Imaris software. This approach has been previously utilized in various cell types including sperm [9,10]. This methodology reduces the target loci to a single point (geometric center) regardless of size or shape of the target; and provides the exact radial localization of each probe in the nucleus by measuring the micrometer distance from the geometric center to the nearest nuclear edge in any direction. Given that the centromere has a much smaller and regular shaped territory than the p- and q-arms, one might expect more variation in the radial position of the chromosomes arms when compared to the centromeres. However, we observed the opposite, with the centromeres exhibiting more variation in radial position than the chromosome arms. We hypothesize that this observation supports the hypothesis that CT position in the sperm nucleus may be determined, at least in part, by the centromeres. This may be particularly true if specific centromeres preferentially form the same chromocenters in sperm nuclei. Previously, we identified an average of seven chromocenters in sperm using both 2D and 3D approaches [10]. To determine whether chromocenters were preferentially comprised of the same chromosomes, we co-hybridized multiple FISH probes and examined their co-localization in 3D with each chromocenter. In this study, we utilized three different FISH probes: (1) a pancentromeric probe, staining all centromeres; (2) a nuclear organizing region (NOR) probe, staining regions located in close proximity to the centromeres of the five acrocentric chromosomes (13, 14, 15, 21, and 22); and (3) a centromeric probe that cross hybridized, staining the centromeres of chromosomes 1, 5, and 19 due to sequence homology. Our findings provided indirect evidence to support the hypothesis that specific chromosomes preferentially clustered to form the same chromocenter. The centromere probe for chromosomes 1, 5, and 19, rarely formed the same chromocenter, and the NOR probe typically resulted in three discrete loci in close proximity to different chromocenters. Additionally, the targeted centromeres and NOR probes rarely formed the same chromocenter, suggesting that chromocenters are not composed of random chromosomes, but may consist of specific chromosomes [10]. Thus, there may be a chromosome-specific composition of individual chromocenters that may reflect the hierarchal organization of chromosomes in the sperm nuclei [6,11,19]. This could potentially account for the larger variability observed for the radial position of the centromeres compared to the p- and q-arms in the current study. Another possible explanation for the smaller variability observed in the radial positioning of the p- and q-arms could simply be due to the much larger target, which is certainly more varied in terms of size, shape, and orientation when compared to individual centromeres (Figure 1 and Figure 2). As the geometric center for each target was used to assess the radial position, it is possible that these factors may reduce the variability of the p- and q-arms compared to the much smaller centromere. 

One of the concerns for the interpretation of the data in this study and study limitations, is the absolute requirement to artificially decondense sperm nuclei prior to FISH due to the extreme compactness of chromatin in sperm [30]. In order, to help maintain the 3D structure of sperm nuclei a mild formaldehyde fixation step was included to induce DNA-protein crosslinks, which has been shown to aid in maintaining the size and shape of nuclei [54]. The agent DTT used to swell sperm nuclei is analogous to disulphide-reducing glutathione, which is found in the cytoplasm of the oocyte, and thus the swelling of the sperm is believed to mimic the decondensation of the paternal genome that occurs following fertilization [29,55]. However, it is critical to note that differences between studies in fixation and swelling protocols in terms of agents and conditions utilized can make study comparisons challenging and could differentially affect CT configurations. For example, visualization of chromocenter(s) in sperm nuclei might be extremely sensitive to decondensation [6]. A handful of studies have assessed chromocenter formation in human sperm by using pancentromeric FISH probes to visualize all centromeres simultaneously. These studies have observed different numbers of chromocenters in human sperm nuclei, with averages varying from 1-to-11 chromocenters in normozoospermic males [6,14,25,31,52]. The decondensation procedure could also potentially affect the configuration and hairpin-loop structure observed in sperm nuclei, which may in part explain different results between studies. Thus, it is critical that decondensation protocols are clearly specified and that decondensation is carefully monitored to cause minimal disruption to the nuclear organization, whilst ensuring robust FISH efficiency. Consequently, it is important to consider that the CT configurations observed in the decondensed sperm nucleus may not entirely reflect the native CT organization in spermatozoa. The fixation and decondensation methodology utilized in this study resulted in minimal decondensation, high FISH efficiency, and reproducible CT organization in five subjects with comparable findings to a previous study and in a different patient cohort [10]. Thus, we are confident that our approach is consistent and at least reflects the nuclear organization of sperm following the decondensation protocol described. 

Differences observed in sperm chromatin organization studies are difficult to resolve and are likely due to inherent differences between studies. For example, there is a large degree of heterogeneity in human sperm samples, and differences may be found between fertile and infertile patients, or may be associated with specific semen parameter disturbances etc. Several studies have reported perturbations, albeit often modest alterations in the positioning of various target loci. Alterations have been reported in patients exhibiting semen parameter disturbances and/or increased sperm aneuploidy [15,19,56,57], increased levels of DNA damage or nuclear perturbations such as sperm nuclear vacuoles [58,59], or carriers of chromosome translocations [45]. There are also inherent methodological differences between studies. These differences include but are not limited to: (1) methodological differences for sperm storage, fixation, and decondensation; (2) differences in the FISH probes utilized; (3) whether 2D or 3D imaging was utilized; and (4) differences in the software and approaches to analyze and determine nuclear organization of probe targets. Thus, direct study comparisons are hampered by these many potential sources of methodological and subject confounders; and are further hindered by the fact that studies often do not adequately describe these important elements to facilitate replication of studies and methodological approaches. 

## 5. Conclusions

Despite the fact, that some published studies are over 20 years old; exploration into the relationship between the organization and function of sperm chromatin organization remains very much in its infancy [6,45]. However, it is clear that spermatozoa possess an evolutionarily conserved unique non-random nuclear organization that differs from somatic cells, suggesting that this organization is functionally significant [16,31]. To date, the function of sperm CT organization remains largely undiscovered [6,44,45]. The current study further supports our hypothesis to suggest that the hairpin-loop model of sperm chromatin organization needs to be refined to reflect a more segmentally organized nucleus. In that, sperm nuclei seemingly do not possess centralized chromocenters, and preferentially form narrower or wider hairpin-loops that can be oriented in any direction in the nucleus, with limited intermingling between chromosome arms. The results suggest that individual chromocenters could be preferentially composed of the same chromosomes, which could determine the segmental radial and longitudinal CT organization, and the hairpin-loop configurations formed. The localization and hairpin-loop configuration of CTs in spermatozoa likely determines the order that CTs or specific genomic regions are exposed to, and remodeled by the oocyte. Thus, nuclear CT position may be critical if there is a functional importance to the order that specific genomic regions are delivered to the oocyte prior to the expression of the paternal genome in the early embryo [6,45]. Therefore, perturbations in sperm CT organization could disrupt the structured events that occur during fertilization, including the formation of the male pronucleus, and early embryonic development [6,14,15,18,29,31,44]. A handful of studies have reported mild to moderate alterations in chromatin organization in the sperm from various patient cohorts including: men with reduced/altered semen parameters, carriers of structural chromosome aberrations, increased sperm aneuploidy and DNA fragmentation etc. [15,19,45,56,57,58,59]. However, these studies are restricted to small numbers of patients and cells. Therefore, despite a relationship between chromatin organization, genome regulation and infertility, it is clear that assessment of chromatin organization is far from ready to be implemented into clinical practice [19]. It is possible that CT organization may be an important marker of chromatin integrity that could identify epigenetic perturbations in infertile men [58]. Future studies are required to establish the function of sperm CT organization and its impact on fertilization and early embryonic development. The relationship between sperm nuclear architecture and genome regulation needs to be deciphered, particularly as it pertains to the progression of spermatogenesis, the formation of mature spermatozoa, fertilization, and post-fertilization events. In other words, is organization of the paternal genome important for the development and function of sperm, formation of the male pronucleus, genome activation, and gene expression in the early developing embryo? Ultimately, this may lead to the development of novel clinically relevant tests to assess epigenetic alterations and fertility potential in men, (albeit unlikely to be assessment of CT organization by FISH). Additionally, this could lead to the development of alternative treatment options to those currently available for the treatment of male infertility. In doing so, we may be able to reduce the financial and emotional burden of infertility and assisted reproductive technology, whilst improving the success rates for those couples trying to conceive.

## Figures and Tables

**Figure 1 genes-10-00504-f001:**
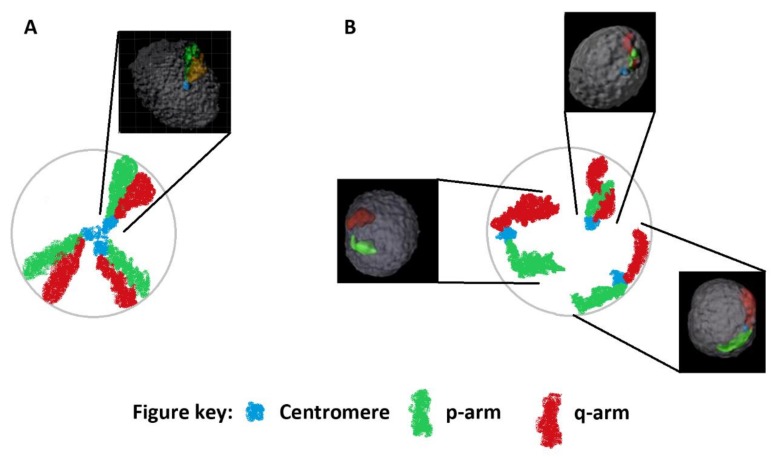
3D rendered models and schematic diagrams illustrating two different models of chromosome organization and hairpin-loop configurations in human sperm. Schematic cross-section of a sperm nucleus, with magnified boxes that display 3D rendered models following sperm-FISH of the organization of centromeres (aqua), p-arms (green) and q-arms (red) in human sperm. Panel A: Displays the original chromocenter hairpin-loop model of sperm chromosome organization, whereby centromeres cluster in the nuclear interior of the sperm nucleus, forming one or more chromocenters, with the p- and q-arms stretching out toward the nuclear periphery. In the current study of 1240 cells, we rarely observed this type of chromosome configuration. This, alongside previously published data [10], led us to propose a refined hairpin-loop model of organization that is more segmentally arranged. Panel B: Segmental model of chromosome organization in human sperm. Here, we observe centromeres distributed throughout the nucleus and the chromosome arms forming narrow and wide hairpin-loops as well as displaying different orientations within the sperm nucleus. Additionally, in contrast to one study [29], we rarely observed chromosome arms to lie in parallel to one another leading to intermingling or coiling of the chromosome arms; with the majority of cells examined displaying discrete non-overlapping territories for the chromosome arms.

**Figure 2 genes-10-00504-f002:**
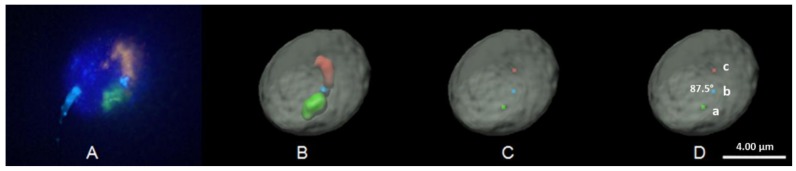
3D FISH imaging and model rendering demonstrating the hairpin-loop configuration for chromosome 2 in a human sperm nucleus. FISH probes utilized corresponded to the centromere (aqua), p-arm (green), q-arm (red) of chromosome 2. The nucleus is counterstained with DAPI (blue: A) and is pseudo-colored (gray: B, C, and D) in 3D rendered models. (**A**) Depicts the raw FISH image following deconvolution (note: aqua signal at position 8 o’clock is background fluorescence from the sperm tail and is removed from 3D rendered models [panels B, C, and D], note the sperm tail was only visible in a small proportion of cells precluding assessment of longitudinal positioning). (**B**) Provides the 3D model reconstruction from the raw deconvolved image (A). (**C**) Depicts the geometric center for each of the three FISH probe targets as determined by the Imaris software. (**D**) Illustrates how the angle of the hairpin-loop configuration is calculated by measuring the angle formed between the p-arm (point a), and the q-arm (point c) through the centromere (point b), which in this cell is 87.5°. It is important to note that the angle created between the chromosome arms is determined in 3D models. In this figure the 3D data is reduced to 2D, which can give the impression that the geometric centers or FISH probes may lie in the same focal plane or z-stack, which is not the case rather this image has compressed the data from 92 sections taken at 0.2 µm intervals, into a single plane. Thus, the angle cannot be accurately assessed in the 2D image and the 3D model has to be utilized to account for differences in FISH probe localizations in different 3D focal planes (X, Y, Z). Scale bar is 4 µm.

**Figure 3 genes-10-00504-f003:**
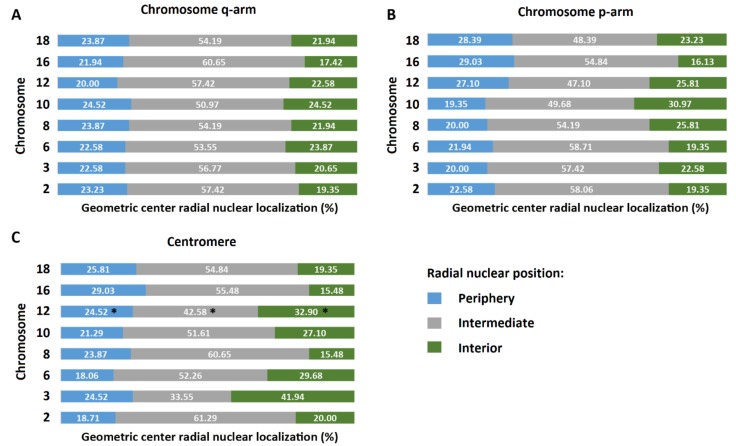
Mean 3D radial positioning of the centromeres, p- and q-arms of eight different chromosomes in human sperm from five normozoospermic subjects demonstrates relative consistent radial organization for the p- and q-arms, with more variation in organization for the centromeres. A minimum of 30 nuclei were examined per FISH probe target, per patient. The data displayed is based on the mean nuclear distribution of the geometric center of each probe targeted in a minimum of a 150 cells in the five patients enrolled in the study. All chromosome centromeres, p- and q-arms were non-randomly distributed in sperm nuclei (*p* ≤ 0.05; χ^2^ goodness-of-fit) with the exception of the chromosome 12 centromere which was randomly organized (*p* ≥ 0.05) *. The nuclear positions of the geometric centers of the q-arms, p-arms and centromeres for each of the eight investigated chromosomes is shown in panels **A**, **B** and **C**, respectively.

**Figure 4 genes-10-00504-f004:**
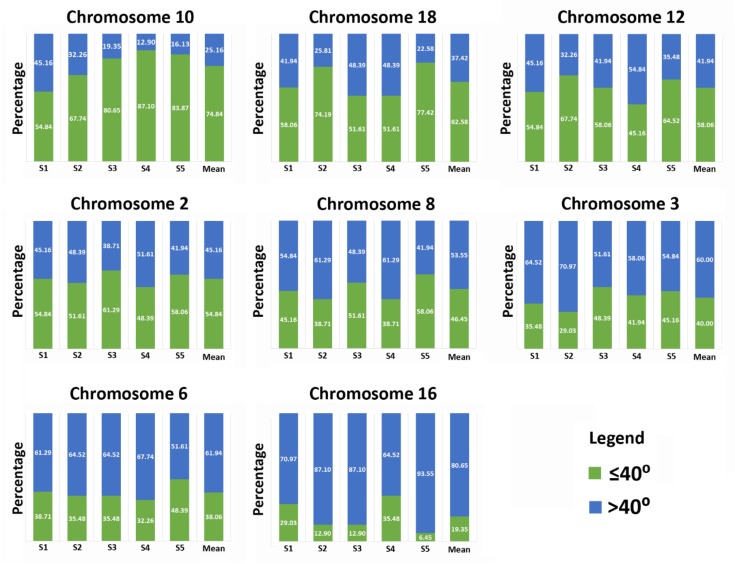
Chromosomes reproducibly have a tendency to form narrower or wider hairpin-loop chromosome configurations in five normozoospermic males for eight investigated chromosomes. The percentage of chromosomes forming narrow (≤ 40°) or wide (> 40°) hairpin-loop configurations for each investigated chromosome is shown for each normozoospermic subject (S1–S5). The mean data for the five subjects is displayed on the far right for each chromosome. Data per subject, per chromosome is based on a minimum of 30 cells; whereas the mean data presented is based on a minimum of 150 cells from the five subjects. Chromosomes are ordered from the top left to the bottom right based on the percentage of narrow (≤ 40°) hairpin-loop configurations for the mean data, with chromosome 6 and 16 preferentially forming the narrowest and widest hairpin-loops, respectively.

**Table 1 genes-10-00504-t001:** Mean micrometer distance of the geometric center of target loci to the nearest nuclear edge in five subjects for eight different chromosomes.

Chromosome	Distance from the Geometric Center of Target loci to the Nearest Nuclear Edge in µm ± SD			
	p-arm	Centromere	q-arm	Sperm Radius (µm)
2	1.87 ± 0.76	1.87 ± 0.73	1.88 ± 0.75	7.72 ± 0.59
3	1.7 ± 0.69	1.48 ± 0.83	1.68 ± 0.74	7.33 ± 0.83
6	1.93 ± 0.74	1.86 ± 0.83	1.91 ± 0.81	7.7 ± 0.94
8	1.83 ± 0.78	1.94 ± 0.76	1.96 ± 0.82	7.96 ± 0.76
10	1.85 ± 0.83	1.9 ± 0.82	1.96 ± 0.83	7.71 ± 0.73
12	1.89 ± 0.86	1.75 ± 0.86	2.0 ± 0.8	7.57 ± 0.68
16	2.18 ± 0.73	1.97 ± 0.72	1.99 ± 0.73	7.52 ± 0.65
18	1.87 ± 0.84	1.83 ± 0.78	1.86 ± 0.82	7.99 ± 0.86

Table displays the mean micrometer (µm) distance to the nearest nuclear edge of the geometric center of the target loci (centromeres, p- and q-arms) for the investigated chromosomes as determined by the Imaris distance transformation tool. The number of determinants for each chromosome target loci is 155 from five different subjects enrolled in the study. The mean µm distance to the nearest nuclear edge is provided with the standard deviation (SD), as well as the average sperm radius for reference.

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
