# Peer review of "Human Sperm Chromosomes: To Form Hairpin-Loops, Or Not to Form Hairpin-Loops, That Is the Question"

_genes, 2019, doi:10.3390/genes10070504_

Round 1

Reviewer 1 Report

            This is an important paper that will add significantly to the discussion of sperm chromatin folding.  The authors are careful to point out the potential difficulties in visualizing sperm chromatin with FISH, namely that the sperm must be decondensed, and to their great credit the authors document how much the nuclei swelled.  This makes the work extremely valuable for the overall discussion on this as yet unsolved problem. 

            Parts of the paper seem very long.  The Introduction could be shortened by half, and the Conclusion should be shortened to a paragraph. 

Line 26 -  Nuclei are more in the range of 10 µm, not 1.

Line 225 -  What is a “1% Tris solution”?  Please just state the concentration of Tris and whether there was any other chemical in this solution.

Author Response

Responses to reviewer 1:

This is an important paper that will add significantly to the discussion of sperm chromatin folding.  The authors are careful to point out the potential difficulties in visualizing sperm chromatin with FISH, namely that the sperm must be decondensed, and to their great credit the authors document how much the nuclei swelled.  This makes the work extremely valuable for the overall discussion on this as yet unsolved problem. 

·        We thank the reviewer for their comments on our work

Parts of the paper seem very long.  The Introduction could be shortened by half, and the Conclusion should be shortened to a paragraph. 

·        We thank the reviewer for their constructive comments we have considerably shortened the introduction and conclusion including deletion of Figure 1, as suggested by reviewer 2. We believe that the manuscript is now much more concise and focused, please see track changes lines 35-210 and Lines 599-681 in the revised document.

Line 26 -  Nuclei are more in the range of 10 µm, not 1.

·        We completely agree and this sentence was meant to reflect micrometers not a single micrometer. See lines 37-38 in the revised manuscript “The linear sequence of our genome is housed in the cell nucleus (~10 micrometers).”

Line 225 -  What is a “1% Tris solution”?  Please just state the concentration of Tris and whether there was any other chemical in this solution.

·        Thank you for pointing out this discrepancy particularly given the need to clearly outline the methodology. This has been updated in the revised manuscript. Lines 240-241. “This was achieved through a 20 minute incubation in decondensation buffer (10mM DTT (Sigma Aldrich, St Louis, MO, USA); 10mM Tris solution, pH 8.0) in the dark.”

Reviewer 2 Report

I have several remarks concerning this paper:

    • The introduction seems 
to be too long
    • In the Introduction, I am missing the explanation of the hairpin-loop configuration model. What do you consider as a hairpin-loop configuration? Is it angle formed by the p- and q-arms through the centromere less than 80 %? This could be described better for each case shown in Figure 2.
    • I don't see how Figure 1 is connected to the organization of sperm DNA
    • The sentence about CTs in line 94 doesn't relay to Figure 1.
    • Line 263: 3. D FISH → 3D FISH
    • You mention that the majority of spermatozoa had chromosome hairpin-loop configurations that were less than or equal to 80°. Why do you use a greater angle as an example in Figure 3?
    • How much 3D organization and chromosome configuration differ among individual sample donors? Some examples of such results would be helpful.
    • What is your explanation of why studies of Mudrak O et al. 2005 and others observed especially hairpin-loop configuration?
    • In the discussion (line 470), there
 are mentioned the negative results concerning the correlation analysis of angle formed and other parameters. Why these results are not mentioned in the Result section or added as a Supplement? Could you analyze the correlation between the angle formed and the distance from the nuclear edge or distance from the connecting piece of sperm tail?  

Author Response

Responses to reviewer 2:

The introduction seems to be too long

·        We thank the reviewer for their constructive comments we have considerably shortened the introduction. We believe that the manuscript is now much more concise and focused, please see track changes lines 35-210.

In the Introduction, I am missing the explanation of the hairpin-loop configuration model. What do you consider as a hairpin-loop configuration? Is it angle formed by the p- and q-arms through the centromere less than 80 %? This could be described better for each case shown in Figure 2.

·        We thank the reviewer for pointing out any lack of clarity. The hairpin-loop model was originally described by the Zalensky group in the 90’s. No strict definition was provided but the term has been adopted within the scientific community. The hairpin-loop term arose based on the groups observation that the centromeres clustered in the nuclear interior and the p- and q-arms stretched toward the nuclear periphery often lying in parallel or intertwined towards the nuclear periphery where the telomeres were clustered forming dimers and tetramers. We have updated the manuscript to provide more description of the model and the adoption of the term. Lines 120-127 “This model described chromosome centromeres clustering in the center of the nucleus to form a single or multiple chromocenters, and the p- and q-arms of each chromosome lying in parallel with each other, or intermingled as they stretched from the center of the nucleus to the nuclear periphery where the telomeres formed dimers and tetramers (Figure 1A)  [25,29-32]. This unique CT configuration was originally described by the Zalensky group as the hairpin-loop model, simply due to its resemblance of a hairpin-loop structure rather than any specific details (e.g., angle formed by the p- and q-arms through the centromere). The description of the hairpin-loop model led to this term being adopted and routinely utilized in the field to describe sperm chromatin organization.”

·        Explanation of the model is also outlined in Figure 1 (old figure 2). Lines 143-146 “Panel A: Displays the original chromocenter hairpin-loop model of sperm chromosome organization, whereby centromeres cluster in the nuclear interior of the sperm nucleus, forming one or more chromocenters, with the p- and q-arms stretching out toward the nuclear periphery.”

 I don't see how Figure 1 is connected to the organization of sperm DNA 

·        Initially this was included to provide more background on CT organization and how still figures neglect to incorporate CTs when describing DNA packaging. However, we recognize this is not central to the general discussion of sperm chromatin organization. As such the figure has been deleted in the revised manuscript and the remaining figures renumbered.

The sentence about CTs in line 94 doesn't relay to Figure 1. 

·        The figure in question has been deleted in the revised manuscript

 Line 263: 3. D FISH → 3D FISH

·        This has been corrected in the revised manuscript and appears to be an error in the conversion to the journal format for review. Line 281.

You mention that the majority of spermatozoa had chromosome hairpin-loop configurations that were less than or equal to 80°. Why do you use a greater angle as an example in Figure 3?

·        There were several reasons to include this specific hairpin-loop configuration: 1) for illustration purposes, as it is much clearer to observe the geometric centers in this type of hairpin-loop configuration than for a narrow configuration thus we believe it allows for a clearer interpretation of how the analysis was performed when compared to a narrow hairpin-loop. 2) Additionally, we wanted to provide examples of various types of configurations observed in Figure 1 and Figure 2 (old Figures 2 and 3 respectively). 3) Additionally, as per the data presented in lines 429-432, the proportion of hairpin-loops >80⁰ was anywhere between 7-38% depending on the chromosome assessed, so we believe that this image is representative of the data.  Lines 429-432 “We stratified chromosomes based on the percentage of cells that contained hairpin-loop configurations that were greater than 80⁰. Between 7-16% of the configurations of chromosomes 2, 8, 10, 12, and 18 were greater than 80⁰, whereas between 25-38% of the configurations for chromosomes 3, 6, and 16 were greater than 80⁰.

How much 3D organization and chromosome configuration differ among individual sample donors? Some examples of such results would be helpful. 

·        We believe that this information is summarized in figure 4 (old figure 5). In figure 4 we present the data of the hairpin-loop configurations in each of the 5 donors and the mean data for each chromosome.

What is your explanation of why studies of Mudrak O et al. 2005 and others observed especially hairpin-loop configuration?

·        Yes this is an interesting question, there are several possible explanations including: 1) the observations of Mudrak were based on fewer and different chromosomes (1, 2 and 5). Additionally, they also were forced to infer centromere position as only the p- and q-arms were probed. Thus, it is possible that for at least chromosome 1 and 5 not included in this study that the hairpin-loop configuration may be tighter as we observe chromosome-specific differences and chromosome 2 is one of the chromosomes we observed to have a relatively tighter configuration. Also differences in the processing of samples, imaging, and analysis may contribute to different findings. Discussion of this is included lines 566-577 “However, it is critical to note that differences between studies in fixation and swelling protocols in terms of agents and conditions utilized can make study comparisons challenging and could differentially affect CT configurations. For example visualization of chromocenter(s) in sperm nuclei might be extremely sensitive to decondensation [6]. A handful of studies have assessed chromocenter formation in human sperm by using pancentromeric FISH probes to visualize all centromeres simultaneously. These studies have observed different numbers of chromocenters in human sperm nuclei, with averages varying from 1-to-11 chromocenters in normozoospermic males [6,14,25,31,52]. The decondensation procedure could also potentially affect the configuration and hairpin-loop structure observed in sperm nuclei, which may in part explain different results between studies. Thus, it is critical that decondensation protocols are clearly specified and that decondensation is carefully monitored to cause minimal disruption to the nuclear organization, whilst ensuring robust FISH efficiency.”     

In the discussion (line 470), there are mentioned the negative results concerning the correlation analysis of angle formed and other parameters. Why these results are not mentioned in the Result section or added as a Supplement? Could you analyze the correlation between the angle formed and the distance from the nuclear edge or distance from the connecting piece of sperm tail?  

·        We thank the reviewer for pointing out this omission. We have included a new paragraph in the results to address this omission. Lines 450-467. “Additionally, correlations between chromosome size, gene density and centromere localization and type of hairpin-loop configurations (narrow or wide) were examined. Analysis of the data presented in Figure 4 shows no obvious correlation with between narrow or wide hairpin-loop configuration and chromosome size. For example, chromosome 16 is the second smallest investigated chromosome and preferentially exhibits a wide hairpin-loop configuration, whereas chromosome 18 is the smallest chromosome, and preferentially exhibits a narrow hairpin-loop configuration. Similarly, no correlation was observed between preference to form narrow or wide hairpin-loop configuration with distance of the centromere to the nuclear edge.Using the Ensembl genome browser correlations between chromosome gene density and hairpin-loop configurations could also be assessed. Chromosome length and number of coding genes for each of the investigated chromosomes was obtained from Ensembl (genome assembly GRCh38.p12). No clear correlation between gene density and tendency to form narrow or wide hairpin loop configurations was observed. For example taking into account the chromosome length and number of protein coding genes, chromosomes 16, 12, 8, 6, 10, 3, 2, and 18 can be organized from highest to lowest gene density respectively. Similar to chromosome size no discernable correlation was observed for gene density and hairpin-loop configuration. For example, looking at the three chromosomes with the highest gene density, they were ranked as third, fifth, and eighth widest hairpin-loop for chromosomes 12, 8, and 16 respectively.”

·        Unfortunately due to lack of background fluorescence we were not able to accurately determine the localization of the sperm tail precluding any correlation with distance to the tail. Lines 331-335 “The unique morphology of the sperm cell often facilitates assessment of longitudinal organization. Unfortunately, in this study the levels of background fluorescence were low, thus, rarely was the sperm tail visible (<10% of analyzed cells) (Figure 2A), which precluded robust assessment of the longitudinal organization of the targeted loci.